# Neural gradients are near-lognormal: improved quantized and sparse training

**Brian Chmiel** [†○∗]  **Liad Ben-Uri** [○∗]  **Moran Shkolnik** [†○]

**Elad Hoffer** [†]  **Ron Banner** [†]  **Daniel Soudry** [○]

[†]Habana Labs – An Intel company, Caesarea, Israel,
[○]Department of Electrical Engineering - Technion, Haifa, Israel

{bchmiel, mshkolnik, ehoffer, rbanner}@habana.ai
{liadgo2, daniel.soudry}@gmail.com

## Abstract

While training can mostly be accelerated by reducing the time needed to propagate neural gradients (loss gradients with respect to the intermediate neural layer outputs) back throughout the model, most previous works focus on the quantization/pruning of weights and activations. These methods are often not applicable to neural gradients, which have very different statistical properties. Distinguished from weights and activations, we find that the distribution of neural gradients is approximately lognormal. Considering this, we suggest two closed-form analytical methods to reduce the computational and memory burdens of neural gradients. The first method optimizes the floating-point format and scale of the gradients. The second method accurately sets sparsity thresholds for gradient pruning. Each method achieves state-of-the-art results on ImageNet. To the best of our knowledge, this paper is the first to (1) quantize the gradients to 6-bit floating-point formats, or (2) achieve up to 85% gradient sparsity — in each case without accuracy degradation. Reference implementation accompanies the paper in the supplementary material.

## 1 Introduction

Neural gradients are used in the training process of deep networks to backpropagate the error-gradient throughout the model, thus allowing to compute the required weight updates. As these neural gradients are needed for a substantial ratio of the underlying computations (about $\frac{2}{3}$), compressing them can alleviate data-throughput requirements and accelerate the training process.

Compression of neural gradients reduce the memory footprint for the intermediate calculation and the bandwidth of data transfer inside the HW accelerator. Moreover, in term of distributed training in model parallelism the neural gradients are one of the main bottlenecks that need to be transferred between devices(Rong et al., 2020; Gupta et al., 2020).

Many previous works (Banner et al., 2019; Fang et al., 2020) compress tensors such as weights and activations by approximating their distributions using an analytically tractable density. These works often assume a bell-shaped distribution such as Gaussian or Laplace distributions, which have been reported to fail for neural gradients (Ye et al., 2019). One key observation in this paper is that neural gradient distributions are heavy-tailed, fundamentally different from the light-tailed distributions of weights and activations. Further statistical and distributional tests reveal gradient magnitudes follow a lognormal distribution.

Adopting this lognormal observation, our paper suggests two main applications — quantization and pruning, used to reduce the computational and memory burden of neural gradients. To tackle these challenges, we first formalize the problems and find closed-form expressions that enable us to predict the optimal quantization and pruning policies. These measures are easy to use and depend only on the estimated lognormal parameters.

---

[∗]Equal contribution.

In Figure 1 we summarize these applications and their derivation. The first application uses the lognormal prior to enabling low-precision floating-point (FP) quantization of the gradients. Here we optimize two tasks. The first task is to find a partition between mantissa and exponent bit-widths that minimizes quantization noise for a given $n$ bit FP gradient representation. The second task is to scale these gradients so that they would be properly represented within a limited dynamic range (distance between the maximum and minimum that FP format can represent). We provide useful insights that make empirically-based heuristics such as *loss scaling* (Micikevicius et al., 2018) a more grounded approach with a theoretical basis. Optimizing both tasks we obtain state-of-the-art results for FP quantization of the neural gradients. The second application performs accurate and predictable stochastic pruning of gradients on the fly, which results in two state-of-the-art pruning schemes. The first translates the desired sparsity level into an accurate threshold, and the other enables combined use of different sparsity levels at different layers (heterogeneous sparsity).

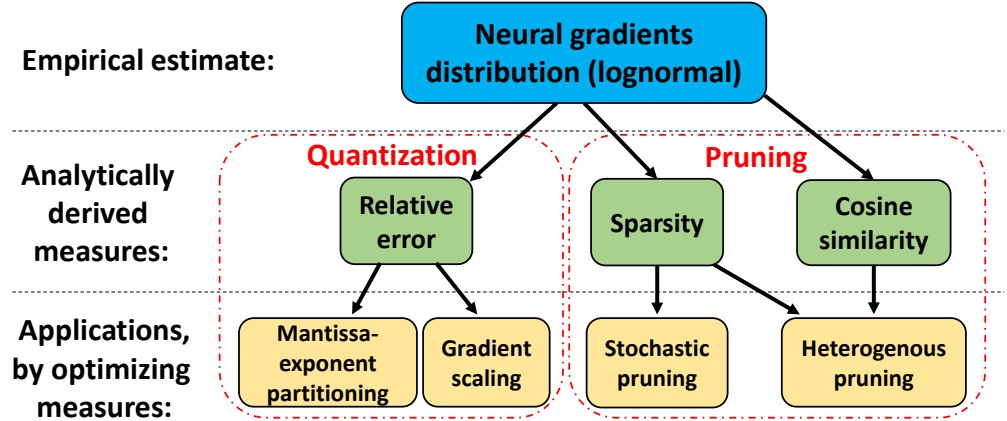

Figure 1: High-level overview of the different methods developed in this work. Adopting the lognormal observation of the neural gradients, we suggest three different measures we can *analytically* optimize: two for pruning and one for quantization. These analytical measures are used in four different schemes.

## 2 RELATED WORK

Quantization and pruning of neural networks have seen a tremendous amount of works (e.g., Nahshan et al. (2019); Choi et al. (2018); Baskin et al. (2018); Louizos et al. (2018); Frankle & Carbin (2018); Cambier et al. (2020)) aiming to reduce both bandwidth and memory footprint, as well as computation time. Most of these methods focus on the quantization or pruning of the weights / activations in the forward path (Banner et al., 2019; Nahshan et al., 2019) or the weight gradients (Bernstein et al., 2018; Alistarh et al., 2016) in the backward path. So far, neural gradients got less attention. Some of these methods (Banner et al., 2019; Ye et al., 2019; Fang et al., 2020) use a systemic and rigorous statistical approach to optimize various distortion measures. For example, Banner et al. (2019) used the normal distributional assumption (of weights and activations) to analytically minimize the mean-squared quantization error. Our work follows a similar line to rigorously optimize similar performance measures for quantization and pruning of gradient distributions, which are different from that of the weights and activations.

**Gradient Quantization.** While a lot of research focused on the quantization of weights and activations for inference (Krishnamoorthi, 2018; Choi et al., 2018; Jain et al., 2020), there were also major advances in quantization during training, many of them faced difficulty trying to represent the high dynamic range of the gradients (Banner et al., 2018; Wu et al., 2018). Cambier et al. (2020) suggests keeping for each tensor shift and scale in full precision numbers to make them fit in FP8 format dynamic range. Mantissa vs exponent allocation of the available bits has proven crucial in deep learning workloads, where for example, BF16 (1-8-7: sign-exponent-mantissa) has shown greater success compared to traditional FP16 (1-5-10) format due to wider dynamic range (Henry et al., 2019; Kalamkar et al., 2019). Research over required format and trade-offs in exponent versus mantissa is on-going with growing interest over lower precision representations such as FP8. Some works have

explored using different FP8 formats: Wang et al. (2018) has used (1-5-2), while (Sun et al., 2019) suggests using one type of FP8 format for the forward (1-4-3), and a different FP8 format (1-5-2) for the backward pass, after empirically assessing the different possible formats. Additionally, with the growing usage of FP16 mixed-precision training, researchers and practitioners faced the need to use loss scaling (Micikevicius et al., 2018), adjusting the tensor distribution to avoid over/under-flow. This procedure required, in some cases, an intensive parameter search and was not guaranteed to succeed in all cases.

**Gradient pruning.** Focusing on the computationally-intensive back-propagation, "meProp" (Sun et al., 2017) prunes the K smallest absolute-valued entries of the neural gradients on the fly, using the top-k algorithm. Works following it, replaced pruning with quantization to induce sparsity (Wiedemann et al., 2020), or used top-k pruning as well on the copies of weights and activations used in back-prop (Aamir Raihan & Aamodt, 2020). Ye et al. (2019), inspired by conventional unbiased estimators like stochastic rounding, suggested "stochastic pruning", reaching higher sparsity levels. Yet, the authors assumed the gradients are normally distributed, leading to incorrect estimation of the threshold and a large difference between the required sparsity and the one obtained. As we shall see later, using the correct statistical distribution model is essential to determine the proper threshold.

## 3 NEURAL GRADIENTS DISTRIBUTION

Many prior works (Banner et al., 2019; Bernstein et al., 2018) take the assumption that tensor data e.g., weights ($W$) and activations ($\mathcal{A}$) is sampled from a Gaussian distribution. Recently, Ye et al. (2019); Wiedemann et al. (2020) used the same assumption for the distribution of neural gradients $\nabla \mathcal{A}$. In the section, we discuss this assumption. We show that neural gradients are better approximated by lognormal distributions, i.e., the gradient logarithm values are normally distributed, as opposed to the gradients themselves.

In Fig. 2, we plot the histogram of neural gradient magnitudes at linear and log scales in one layer of ResNet18 - ImageNet dataset. At a linear scale, the distribution has a few gradients of huge magnitude and a great many gradients of small magnitudes (Fig 2a). Plotting the histogram on a logarithmic scale reveals a distribution close to a symmetric normal (Gaussian) distribution (Fig 2b). This is the hallmark of the lognormal distribution. Finally, when plotting the theoretical quantiles of the normal distribution against the quantiles of the gradient distribution (Q-Q plot), we see that the points follow a strongly nonlinear pattern in Fig 2c, suggesting that the data is not distributed as a standard normal distribution. Note that in the Q-Q plot for lognormal distribution (Fig 2d), almost all points lie on a straight line.

We further estimate the goodness of fit of the neural gradients to normal and lognormal distributions. To that end, we measure the static distance (largest vertical line) between the cumulative distribution function (CDF) of the empirically observed distribution and the CDF of the reference distribution (also known as Kolmogorov-Smirnov test (Smirnov, 1948)). For each model and dataset in Table 1, we calculate the average (across all layers) of the static distance to normal and lognormal distributions. The analysis is performed on the absolute value of the gradients, excluding the zero-valued entries. Note that lognormal distribution gets a better fit. Additional statistical distributions in Appendix A.1 .

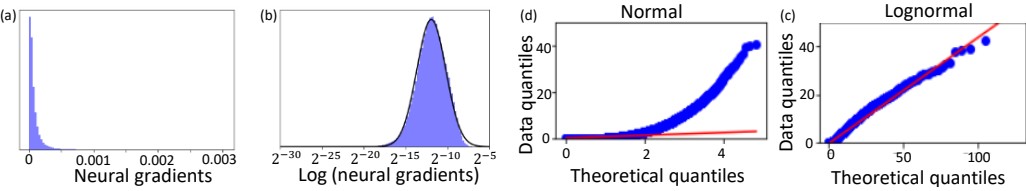

Figure 2: Identifying the distribution of neural gradients (normal vs. lognormal): **(a)** probability density function of gradient magnitudes; **(b)** the same histogram on a logarithmic scale — notice that the shape looks very much like the symmetrical shape of the regular normal distribution; **(c)** quantiles of gradient magnitudes against the quantiles of a normal distribution; and **(d)** quantiles of gradient magnitudes against the quantiles of a lognormal distribution; Additional layers' distribution in Fig. A.2. In order to emphasize the difference between neural gradients and weights gradients we show the quantiles of weights gradient in Fig. A.23

Table 1: Mean $\pm$ std (p-value) over all layers over time of KS test on different models and datasets for normal and lognormal distribution. Notice the lognormal distribution gets the higher fit across all models. We compare to additional distributions such as Laplace, uniform, Cauchy and loglaplace in Appendix A.1 Also, in Table A.2 we show there is a good match to the lognormal through training (not only on average), and in Fig. A.3 we show this holds also for specific layers over time .

| Distribution | Model (Dataset) | | | | | |
|---|---|---|---|---|---|---|
| | BERT (CoLa) | BERT (MRPC) | ResNet18 (ImageNet) | MobileNetV2 (ImageNet) | VGG16 (ImageNet) | DenseNet121 (ImageNet) |
| Normal | $0.46\pm0.02$ $(2\cdot 10^{-4})$ | $0.39\pm0.04$ $(5\cdot 10^{-5})$ | $0.38\pm0.1$ $(3\cdot 10^{-6})$ | $0.22\pm0.09$ $(5\cdot 10^{-6})$ | $0.35\pm0.08$ $(3\cdot 10^{-6})$ | $0.33\pm0.1$ $(5\cdot 10^{-5})$ |
| Lognormal | $0.05\pm0.002$ $(0.28)$ | $0.04\pm0.002$ $(0.23)$ | $0.02\pm0.002$ $(0.26)$ | $0.07\pm0.003$ $(0.18)$ | $0.06\pm0.002$ $(0.31)$ | $0.05\pm0.001$ $(0.29)$ |

## 4 APPLICATION I - OPTIMAL FLOATING-POINT QUANTIZATION

Floating-point (FP) representations can cover a very wide dynamic range with a relatively small number of digits. The mantissa-exponent tradeoff control the balance between the dynamic range of numbers that can be represented (exponent) to the precision of these numbers (mantissa). This dynamic range is especially important for the heavy-tailed distributions that characterize neural gradients. In this section, we study the optimal characteristics of the FP quantizer.

### 4.1 PROBLEM FORMULATION

We can decompose any positive real value $x \in \mathbb{R}^+$ as follows:

$$x = 2^{\ln x} = \overbrace{2^{\ln x - \lfloor \ln x \rfloor}}^{M \in [1,2)} \cdot \overbrace{2^{\lfloor \ln x \rfloor}}^{E \in \mathbb{Z}}, \tag{1}$$

where $M \in [1, 2)$ is the mantissa and $E \in \mathbb{Z}$ the exponent. Given $N$ bits, we allocate 1 bit for the sign and seek the optimal allocation of $n_1$ bits to $M$ and $n_2$ bits to $E$, such that $n_1 + n_2 = N - 1$. Accordingly, we define the quantized $x_q$ as:

$$x_q = \begin{cases} 2^{E_{\max}} & E \geq E_{\max} \\ M_q \cdot 2^E & -E_{\max} \leq E \leq E_{\max} \\ 0 & E \leq -E_{\max} \end{cases} \tag{2}$$

where $E_{\max} = 2^{n_2 - 1}$ and $M_q$ is the quantized mantissa with the range $[1, 2)$ divided into $2^{n_1}$ quantization levels with a spacing of $\Delta = \frac{1}{2^{n_1}}$.

Finally, we measure the relative error between the FP number $x_q$ and the real number $x$, which is simply the difference between the two numbers divided by the real number (Widrow & Kollár, 2008):

$$\eta(n_1, n_2) = \left| \frac{x_q - x}{x} \right| \tag{3}$$

### 4.2 ANALYTICAL DERIVATION OF THE RELATIVE ERROR

We assume that $x \sim \text{Lognormal}(\mu, \sigma^2)$. Note that $E = \lfloor \ln x \rfloor \approx \ln x \sim \mathcal{N}(\mu, \sigma^2)$. In Appendix A.4 we split the range into three parts according to $E$: (i) $-E_{\max} \leq E \leq E_{\max}$; (ii) $E \geq E_{\max}$; (iii) $E \leq -E_{\max}$, and calculate the expected contribution for each term. A closed-form formula for the expected relative error could be obtained as follows:

$$\begin{aligned} E\left[\eta(n_1, n_2)\right] = & \frac{2\Phi\left(\frac{E_{\max}}{\sigma}\right) - 1}{8 \cdot \ln(2) \cdot (2^{n_1} - 1)} + 2^{E_{\max} - 1} e^{\frac{\sigma^2 \ln^2(2)}{2}} \left( \text{erf}\left( \frac{\sigma \ln 2}{\sqrt{2}} + \frac{E_{\max}}{\sqrt{2}\sigma} \right) - 1 \right) \\ & - \frac{1}{2} \text{erf}\left( \frac{E_{\max}}{\sqrt{2}\sigma} \right) + \frac{3}{2} - \Phi\left( \frac{E_{\max}}{\sigma} \right) \end{aligned} \tag{4}$$

where $\Phi(x)$ is the CDF of $\mathcal{N}(0,1)$. In Fig. 3a we show that analytical results stated by Eq. (4) are in good agreement with simulations for FP8 with various number of exponent bits. Simulations were obtained by quantizing 10,000 values, generated from a lognormal distribution with $\sigma = 1, 3, 5$.

### 4.3 THE OPTIMAL MANTISSA-EXPONENT REPRESENTATION

The relative error in Eq. 4 depends on the scale parameter $\sigma$, the number of bits of the mantissa $n_1$ and exponent $n_2$, respectively (the latter through $E_{\max} = 2^{n_2-1}$). Given any $N$-bit FP format, we wish to find a mantissa-exponent partition that minimizes the expected relative error such that $n_1 + n_2 = N - 1$. Minimizing Eq. 4 yields this optimal partition. To do so we set $n_1 = N - n_2 - 1$, equate the derivative to zero and solve. The computational cost of such a solution is negligible (details are in Appendix A.4.4). This allocation depends on $N$ and $\sigma$. In Fig. 3b we show the optimal allocations for $N = 5, 6, 7, 8$ bits and $\sigma \in [1, 8]$. In Fig. 3c we show the FP format obtained by solving Eq. (3) with normal distribution assumption for neural gradients, which leads to a sub-optimal format as shown in Table 3. The full solution can be found in Appendix A.5.

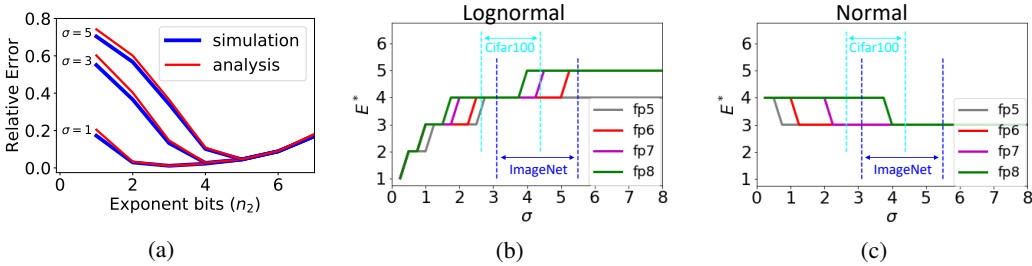

|  |  |  |
|:---:|:---:|:---:|
| (a) | (b) | (c) |

Figure 3: **(a)** Expected relative error as a function of $n_2$ for FP8 with a lognormal distribution. The simulation is in good agreement with the analytical results stated by Eq. (4). **(b)** Ideal bit allocation for the exponent ($E^*$) as a function of $\sigma$, using Eq. (4). Notice that $\sigma$ varies between datasets, imposing different optimal allocations. **(c)** Ideal bit allocation assuming a normal distribution, which leads to wrong FP formats, e.g., the number of bits assigned to the exponent decreases when $\sigma$ (dynamic range) increases.

Table 2 summarizes the statistical analysis applied for various floating point formats. Empirical observations prove that gradient distributions have a range of $[3, 5.5]$ and $[2.5, 4.5]$ in ImageNet and Cifar100 datasets respectively, which we use to determine the optimal mantissa-exponent partition for FP4–FP8. In Section 6, we use these partitions to train ImageNet and Cifar models at reduced precision (bit-width lower than 8), and show empirically that these partitions provide the best results.

Table 2: Ideal sign-exponent-mantissa bit allocations based on the optimization of Equation 4 and the gradient statistics of ResNet18, ResNet101 and SqueezeNet in Cifar100 and ImageNet datasets. The analysis matches the previously suggested allocation by Sun et al. (2019) for FP8, showing a clear benefit of 1-5-2 for gradient quantization.

| Models | Dataset | $\sigma$ Range | FP4 | FP5 | FP6 | FP7 | FP8 |
|:---:|:---:|:---:|:---:|:---:|:---:|:---:|:---:|
| ResNet18, ResNet101 | Cifar100 | 2.5-4.5 | 1-3-0 | 1-4-0 | 1-4-1 | 1-4-2 | 1-5-2 |
| ResNet18, SqueezeNet | ImageNet | 3-5.5 | 1-3-0 | 1-4-0 | 1-5-0 | 1-5-1 | 1-5-2 |

### 4.4 PER-LAYER GRADIENT SCALING

The use of loss scaling (Micikevicius et al., 2018), is key to the quantization of gradients using low precision FP formats. The idea is to shift the neural gradients' dynamic range to fit the floating-point range thus avoiding possible underflows. Loss scaling is usually performed by multiplying the loss value by a large constant and then dividing weight gradients by the same value after back-propagation and before any update has taken place. As gradient distribution changes across training, dynamic loss scaling is sometimes needed. In this setting, gradients are monitored for any overflow or underflow

that may occur. Upon this occurrence, gradients are discarded for that step and loss scale value is changed to combat the phenomena (either increased or decreased heuristically).

Choosing either a fixed or dynamic *global* loss scale that can fit across all the layers is challenging and may prove impossible for low precision FP formats that further reduce the dynamic range. In Fig. A.5, we show the standard deviation $\sigma$ of the gradient at the log scale for different transformer layers. The high variability in $\sigma$ between the layers makes the choice of one global loss scale unlikely. The variation in gradient statistics across layers may also explain the need for previous hybrid approaches that kept some layers at higher precision. For example, Sun et al. (2019) reported that when training ResNet18 models using FP8, they had to keep the first and last layers at FP16 precision. Fig. A.6 suggests a reason — the first and last layers exhibit std that is very different from the other layers (i.e., they require a different gradient scaling ). Cambier et al. (2020) showed the low results achieved using a global loss scale in FP8 format in transformer and suggested a full precision (expensive) computational operation of rescaling and shifting the neural gradients to FP8 dynamic range.

We therefore suggest to use a per-layer *gradient scale*, instead of a global loss scale. As detailed in Appendix A.6, our gradient scaling method keeps the largest gradients representable but sacrifices the smallest gradients. These tiny gradients can be pruned without significantly distorting the original tensor because: (1) the lognormal distribution suggests that gradients of tiny magnitude are relatively scarce; (2) such tiny gradients are typically less significant in training compared to the larger gradients. Pseudo-code appears in Algorithm 1

## 5 APPLICATION II - STOCHASTIC PRUNING

Inspired by conventional unbiased estimators for quantization such as "stochastic rounding", researchers have recently proposed "stochastic pruning" (Ye et al., 2019), an unbiased pruning method that introduces zero bias on expectation.

Given a threshold $\alpha$, we sample a uniform variable $\varepsilon \sim U[0, 1]$ and prune $x$ as follows:

$$T_{\alpha,\varepsilon}(x) = \begin{cases} x & |x| > \alpha \\ \text{sign}(x) \cdot \alpha & \alpha \cdot \varepsilon \leq |x| \leq \alpha \\ 0 & |x| < \alpha \cdot \varepsilon \end{cases} \qquad (5)$$

The method is graphically illustrated in Figure 4. Note that all values in the range $[-\alpha, \alpha]$ need to be mapped to only one of the three possible values $(0, \pm\alpha)$. Their increased frequency of occurrence can be used to design a custom encoding method to compress their representation. In Appendix A.10, we propose an encoding method with a compression ratio equivalent to quantizing to 4 bits at 80% sparsity or to only 2 bits at 90% sparsity.

### 5.1 PROBLEM FORMULATION

Our goal would be to find an analytical expression for a proper threshold $\alpha$ that induces the desired sparsity $S$ using stochastic pruning. Specifically, let $x$ be a random variable with a known distribution and $S$ a given sparsity level ($0 < S < 1$). Using stochastic pruning, with $\varepsilon \sim U[0, 1]$, we obtain:

$$S = \mathbb{E}_\varepsilon \int_0^{\alpha \cdot \varepsilon} f(x) \, dx \qquad (6)$$

### 5.2 ANALYTICAL DERIVATION OF SPARSITY

Understanding the lognormal distribution of the gradients, Eq. (6) is simply the CDF of a lognormal distribution at $\alpha \cdot \epsilon$, that is:

$$S = \mathbb{E}_\varepsilon \left[ \frac{1}{2} + \frac{1}{2}\text{erf}\left( \frac{\ln(\alpha \cdot \varepsilon) - \mu}{\sqrt{2}\sigma} \right) \right] \underset{\tau = \frac{\varepsilon}{e^\mu}}{=} \int_0^{\frac{\alpha}{e^\mu}} \left[ \frac{1}{2} + \frac{1}{2}\text{erf}\left( \frac{\ln(\tau)}{\sqrt{2}\sigma} \right) \right] \frac{e^\mu}{\alpha} \, d\tau \qquad (7)$$

The complete solution for this integral can be found in Appendix A.3, resulting in:

$$S = \frac{1}{2} + \frac{e^\mu}{2\alpha} \left[ e^{\frac{\sigma^2}{2}} \text{erf}\left( \frac{\sigma}{\sqrt{2}} - \frac{\ln\left(\frac{\alpha}{e^\mu}\right)}{\sqrt{2}\sigma} \right) + \frac{\alpha}{e^\mu} \cdot \text{erf}\left( \frac{\ln\left(\frac{\alpha}{e^\mu}\right)}{\sqrt{2}\sigma} \right) - e^{\frac{\sigma^2}{2}} \right] \qquad (8)$$

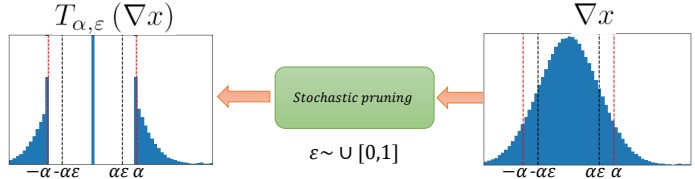

Figure 4: Effect of stochastic pruning on a lognormally distributed tensor. The threshold can be found from Eq. (6) according to the gradients' distribution. $S$ is the ratio of values mapped to 0, notice that a large fraction of values, all values $\alpha \cdot \varepsilon \leq x \leq \alpha$ are mapped to $\pm\alpha$ — two values that can have a special encoding, thus reducing the tensors' memory footprint — details in Appendix A.10.

Eq. (8) can easily be solved numerically to find $\alpha$. As shown in Fig. A.12 the lognormal parameters $\mu$ and $\sigma$ of the gradients' distribution at each layer are pretty stable throughout the training, which allows sampling $\sigma, \mu$ and calculate the threshold $\alpha$ not very frequently. In practice, we perform this procedure only once per epoch while achieving stable sparsity levels throughout the training. Moreover, the computational complexity of solving Eq. (8) is negligible (empirically it converges in a few iterations); further details are found in Appendix A.7.

## 5.3  HETEROGENEOUS SPARSITY ALLOCATION

We show in Fig. A.9a that the angle between the tensors before and after the stochastic pruning (measured by the cosine similarity) can serve as an important proxy to the overall validation accuracy achieved. Interestingly, using the cosine similarity, we observed that stochastic pruning takes a different toll from the different layers, i.e. pruning all layers to the same sparsity level damages some of them more than others. This phenomenon can be explained and assessed by analyzing the cosine similarity of a lognormal distribution, where the difference between the layers is the parameters of the distribution. We derive the cosine similarity as another analytical measure and propose an algorithm that better preserves the cosine similarity of some of the layers, by decreasing their sparsity level, while increasing the sparsity level of other layers — maintaining the overall sparsity budget (mean sparsity of all the layers). Further details can be found in Appendix A.9. This allows us to increase the sparsity level while preserving accuracy, results can be seen in Section 6.

## 6  EXPERIMENTS

In this section, we evaluate the methods and predictions above, all stemming from the lognormal distribution of the neural gradients, for the two suggested applications: floating point format quantization and stochastic pruning. Experiments, details, and additional results appear in Appendix A.12.

**Floating point format.**  In Table 3 we show the results of different allocations between exponent and mantissa for different FP formats in Cifar100 and ImageNet dataset. We quantize all convolutional layers' gradients, unlike previous methods (Sun et al., 2019) that keep part of them in FP32. All results were achieved using the suggested gradient scaling, where the mean is sampled once every epoch. For all FP formats, the results fit the analysis in Table 2. In contrast, we get sub-optimal FP format if instead we solve Eq. (3) with a normal distribution assumption.

**Per-layer gradient scaling**  In Table 4 we compare the suggested gradient scaling with static and dynamic (Micikevicius et al., 2018) global loss scaling. We clipped the values to maximum/minimum FP representation for our method and the 'static' method to avoid overflow/underflow, respectively. On the other hand, the dynamic method (Micikevicius et al., 2018) clips the values to adjust the scale in response to overflow/underflow in the weights updates. Our suggested layer-wise gradient scaling achieves better results, highlighting that one global loss scale for the entire network might be too restrictive in practice. In Fig. A.7, we show the variability of the suggested gradient scale across different layers. Lastly, to verify that gradient quantization is the main bottleneck, we also quantized the weights and activations to INT4 and found a minor degradation in accuracy (0.3%).

Table 3: Floating-point formats for neural gradient representations. The table reports the validation accuracy of ResNet18 on ImageNet & Cifar100, ResNet101 on Cifar100 and SqueezeNet on ImageNet. $E^*$ denotes the optimal exponent bit-width, which minimizes the expected relative error of Eq. (4). N/A refers to cases of invalid format (e.g., 1-5-0 is the optimal FP6 format for ImageNet, but FP6 cannot have $E^* + 1 = 6$ exponent bits). Note that in all cases, the formats that are predicted by Eq. (4) to minimize the relative error also provide the highest accuracies (**bolded**). We marked (by $^\dagger$) the results for the optimal format assuming normal distribution for neural gradients in Eq. (3).

| Dataset | Model | $\sigma$ Range | Baseline | FP | $E^*$ | $E^*$+1 | $E^*$-1 | $E^*$-2 |
|---------|-------|---------|----------|-----|-------|---------|---------|---------|
| Cifar100 | ResNet18 | 2.5 - 4.5 | 64.9% | FP5 | **64.0%** | N/A | 58.9%$^\dagger$ | 26.6% |
| | | | | FP6 | **64.9%** | 64.6% | 59.7%$^\dagger$ | 28.6% |
| | ResNet101 | 2.5-4.5 | 71.3% | FP5 | **70.4%** | N/A | 66.5%$^\dagger$ | 35% |
| | | | | FP6 | **70.97%** | 70.82% | 67.5%$^\dagger$ | 42.7% |
| ImageNet | ResNet18 | 3 - 5.5 | 70.4% | FP6 | **70.0%** | N/A | 67.1%$^\dagger$ | 30.8% |
| | | | | FP7 | **70.4%** | 70.1% | 66.7% | 47.5%$^\dagger$ |
| | SqueezeNet | 3 - 5.5 | 58.19 % | FP5 | **55.2%** | N/A | 47.3%$^\dagger$ | 33.2% |
| | | | | FP6 | **57.8%** | N/A | 56.1%$^\dagger$ | 54.3% |

Table 4: Comparison of the suggested gradient scaling against different global fixed loss scaling and global dynamic loss scaling (Micikevicius et al., 2018) in ResNet18, on the ImageNet dataset for the optimal format in FP4 (1-3-0).

| Baseline | Gradient scaling (Ours) | Dynamic Global | Static (Global) | | | | | | |
|----------|----------|---------|----------|----------|----------|----------|----------|----------|----------|
| | | | $2^{13}$ | $2^{14}$ | $2^{15}$ | $2^{16}$ | $2^{17}$ | $2^{18}$ | $2^{19}$ |
| 70.4% | **64.8%** | 3% | 26.7% | 26.4% | 38.7% | 41.5% | 42.2% | 54.9% | 53.9% |

**Stochastic pruning.** In Section 6 we compare the proposed methods for homogeneous and heterogeneous stochastic pruning against SAW (Aamir Raihan & Aamodt, 2020) that uses "top-k" pruning and ATP (Ye et al., 2019) that also uses stochastic pruning but assumes the gradients are normally distributed. Notice that stochastic pruning outperforms the non-stochastic top-k, and that the heterogeneous version surpasses the homogeneous one. The validation accuracy during training for different sparsity levels and different datasets can be found in Fig. A.16. In Section 6 we demonstrate our methods' ability to produce the required sparsity level, for both the homogeneous and heterogeneous versions. In contrast, the sparsity is not accurate for the baseline methods: (1) finding the threshold using top-k and then applying stochastic pruning, and (2) using ATP (Ye et al., 2019), which assumes a normal distribution. In Fig. A.15 we see how the sparsity inaccuracy occurs at all layers, and in Fig. A.1a we see how other distributions (not lognormal) cause an inaccuracy. This strengthens the importance of using the correct distribution of the neural gradients in Eq. (6).

# 7 SUMMARY

We evaluated the distribution of neural gradients and showed they can be well-approximated as a lognormal distribution. We use this distribution to analytically derive accurate measures (e.g., sparsity and local distortion metrics), useful for the two following applications:

**Quantization.** We found the optimal bit allocation to the mantissa and exponent for a floating-point gradient representation, explaining prior results for FP8 and paving the way towards lower accuracy representations. We suggest using a per-layer gradient scale and find its optimal value, preventing under/over-flow in scenarios that challenged prior methods or required an intensive parameter search. Combining both methods, we trained using low precision neural gradients on ImageNet and achieved, for the first time, no noticeable validation accuracy degradation with FP7 and FP6.

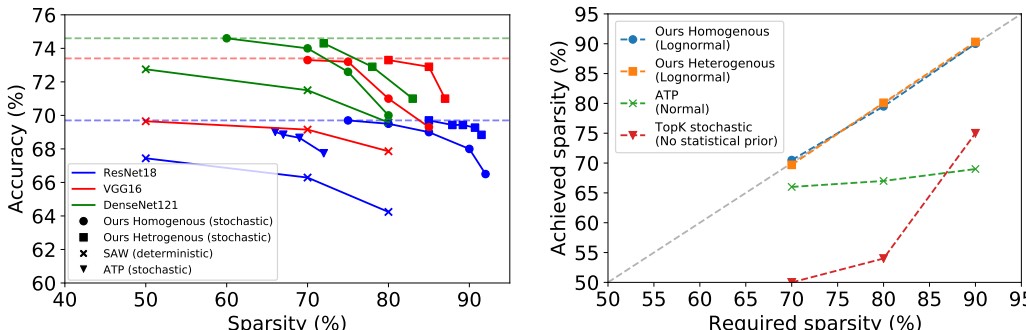

Figure 5: **(Left)** Comparison of the homogeneous and heterogeneous stochastic pruning methods against pruning with SAW (Aamir Raihan & Aamodt, 2020), which uses the "top-k" method and ATP (Ye et al., 2019), which also uses stochastic pruning, but assumes the neural gradients are normally distributed. These are compared on 3 different architectures (ResNet18, VGG16 and DenseNet121). Our proposed methods achieve higher sparsity while maintaining baseline validation accuracy. **(Right)** Comparison of the required and achieved sparsity of our method, ATP (Ye et al., 2019) and applying "top-k" followed by stochastic pruning for ResNet-18 (additional models in Fig. A.24). The validation accuracy for our method is never less than the accuracy of the other methods for the same achieved sparsity. Notice the large deviation in the other methods between the achieved and required sparsity. This emphasizes the importance of using the correct distribution in order to both enjoy the improved performance of stochastic pruning over regular "top-k" and maintain the ability to fully control the achieved sparsity. Additional details and results are in Appendix A.12

**Pruning.** We can use stochastic pruning to prune the neural gradients during training precisely to a predetermined sparsity level, with minimal overhead computation. We show that this form of stochastic pruning is superior to deterministic pruning. Specifically, we have achieved up to 80% gradient sparsity without hurting validation accuracy (ResNet18 on ImageNet) using a homogeneous sparsity level for all layers. We also show that the uniform sparsity method is sub-optimal with respect to an analytical error measure we derive (the cosine similarity), and suggest allocating different sparsity levels to the different layers to preserve it better. We suggest and test an algorithm for this allocation - allowing for more aggressive overall pruning, achieving 85% gradient sparsity while preserving baseline accuracy, and reaching nearly 90% with less than 0.3% accuracy degradation.

**Why we get lognormal distribution?** From the central limit theorem, normal distributions universally arise from the sum of many random variables, while lognormal distributions universally arise as the product of many random variables. This might suggest that, as we backpropgate the neural gradients through the network, these gradients have a few dominant paths, so most significant operations are products, rather than summations (i.e., so along these paths: effective depth $\gg$ effective width). Recent work (Hanin & Nica, 2018) made this explanation rigorous at initialization, in the limit where both the width and depth of the neural network jointly go to infinity. However, we suspect this explanation is only a part of a more nuanced picture. Understanding this is an interesting direction for future research.

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
