# OpenReview forum: "Neural gradients are near-lognormal: improved quantized  and sparse training"
_ICLR.cc/2021/Conference — ICLR 2021 Poster_

### Official Review · AnonReviewer2 · 2020-10-27

**Rating:** 7
**Confidence:** 4

**Review:**

Following the author response, I have updated my score to an accept. The key contribution is likely to be of interest to the ICLR community and the paper is well executed.

---

The paper proposes a principled approach to the compression of intermediate gradients by fitting a log-normal distribution to the individual derivatives, which they show is a good fit for the studied models. Although simple a posteriori, the proposed approach is elegant and the arguments are presently clearly and well illustrated.

The weak points of the submission are a lack of clarity for some of the assertions and a lack of motivating example or application. The current writing takes for granted that quantizing intermediate gradients is good. The claimed benefits of quantization is to "alleviate data-throughput requirements and accelerate the training process"
and "reduce both bandwidth and memory footprint, as well as computation time". Those benefits are more obvious in a distributed setting, where the main bottleneck to training is the communication of gradients. This reasoning does not directly apply to intermediate gradients. Quantization and low-prevision might very well provide those benefits in the non-distributed setting, but the current exposition does not make it concrete, which might limit its impact.

My initial recommendation is towards acceptance if those points and some smaller points below can be adressed during the discussion period. The submission is well executed and its contribution is likely to be of further theoretical and practical interest.

Smaller points;
- The introduction mentions that 2/3 of the underlying computation involve intermediate gradients. Please make this number concrete through an example, as this seems highly dependent on the architecture and data type.
- One of the main points of the introduction is that the intermediate gradients are fundamentally different from weight gradients. This assertion should be supported by data in the submission, maybe as an addition to Figure 2.
- The term `neural gradient` being undefined in the abstract and title is an issue. While its meaning can be inferred, it is not standard terminology and needs to be defined in the abstract or replaced.
- The term `dynamic range` would benefit from a technical definition.




Minor points, not part of the main decision
- Something went wrong in Equation (7).  $\tau$ seems to be used instead of what was originally planed as $\epsilon'$?
- Some `\refs` seem broken in Section 4.3 as they point _to_ section 4.3
- The acronym `BF16` does not seem defined in text.
- The value of the Kolmogorov–Smirnov test and Table 1 is limited by a lack of interpretability, since it is not aimed at a p-value. Adding more candidate distributions to communicate the range of possible values would help.
- A short definition of the tradeoff between mantissa and exponent might help a general audience.
- Some of the references point to preprint versions when the work has been published, for example [Cambier et al. 2020], or does not give appropriate bibliographical information, for example [Choi et al. 2018a]. Please take the time

---

> ### Author Response · Authors · 2020-11-18
> **Authors response for Reviewer #2**
>
> 1) The compression of neural gradients reduces bandwidth, memory footprint and computational time. The bandwidth and memory footprint is reduced since in the backpropagation process all the intermediate calculations are saved in the device memory before the calculation of next layers (usually cache memory can't contain all these intermediate calculation and it requires to be transferred to SRAM).  Additionally, in distributed training of model parallelism, the neural gradients may form a bottleneck, as these gradients are required to transfer across devices (see recent paper by Facebook [1] explaining this problem in recommendation systems).  We agree with the reviewer that when talking about distributed training with data parallelism, main bottlenecks relate to the weight gradients, where there are a variety of of works that try to confront with this problem ([2,3]).  Computational effort is also reduced, since the neural gradients are part of the calculation of next layer and weight gradients at backpropagation.
>
> We added a new paragraph to the introduction explaining this motivation.
>
> 2)  The computations of any neural network contain 3 different parts: The calculation of the activations ($Y = \sigma(W*X)$) in the fwd pass, the calculation of the weights gradients (dL/dW) and the calculations of the neural gradients (dL/dY). The neural gradients are needed for the calculation of the last two.
>
> 3) As proposed by the reviewer we added to the appendix Fig A.23 which is similar to Fig.2, but for weight gradients. It is noticeable that the weights gradients don't follow lognormal distribution, but are closer to a normal distribution.
>
> 4)  In updated version, we moved the explanation of the term neural gradients ("gradients with respect to intermediate neural layer outputs") from the introduction to the abstract.
>
> 5)  We added an explanation of the term "dynamic range" the first times it appears in the introduction.
>
> 6) We fixed equation 7.
>
> 7)  The refs in section 4.3 were fixed.
>
> 8) BF16 refers to BFloat16 format (1-8-7 bits for the sign-exponent-mantissa respectively). Added explanation in the text.
>
> 9) We added to table1 the p-values. Also notice in table A.1 in the appendix you can find more candidate distributions - Showing lognormal is a better fit for neural gradients distributions..
>
> 10)  We added a definition of the mantissa-exponent trade-off in the first paragraph of section 4.
>
> 11) We fixed the references in the updated version
>
>
> [1] Gupta, Vipul, et al."Fast Distributed Training of Deep Neural Networks: Dynamic Communication Thresholding for Model and Data Parallelism". https://arxiv.org/pdf/2010.08899.pdf
>
> [2] Alistarh, Dan, et al. "QSGD: Communication-efficient SGD via gradient quantization and encoding." Advances in62Neural Information Processing Systems. 2017
>
> [3] Bernstein, Jeremy, et al. "signSGD with majority vote is communication efficient and fault tolerant." arXiv preprint64arXiv:1810.05291 (2018).

---

### Official Review · AnonReviewer3 · 2020-10-29
**interesting paper for suggesting a better distribution of gradients, but more details and practical implications should be covered**

**Rating:** 7
**Confidence:** 3

**Review:**

The authors suggest an interesting finding where the gradient distribution in each layer is close to log-normal distribution instead of normal distribution. Given the suggestion, the authors propose two closed-form analytical methods to produce a better low-precision floating-point format and an optimized sparsity threshold for gradient pruning. While such distribution assumption with associated analytical methods seems promising, there are a few suggestions and questions I want to address for the paper:

1. There are many previous studies on quantizing gradients for efficient distributed training, such as QSGD [1] and signSGD [2]. Although most of those studies do not reduce the computational complexity for backpropagation, they still serve as well-performed gradient quantitation methods. I think it's better to distinguish from those previous studies explicitly in the paper.

2. It's better to involve more motivations and explanations about why gradient quantization is important for reducing the computational and memory burden of neural gradients.

3. In the context of the real hardware design, it would be hard to get a computational reduction by using the proposed method which determines the optimal floating-point representation. Real hardware design (e.g. GPU architecture) requires a global floating representation across all datasets so it is impossible to design hardware with a specific floating representation for a specific dataset. Therefore, I don't think determining the best floating-point format for each dataset is useful in practice.

4. Also I would like to address some technical questions regarding the setting and experiment:

a. How does weight update work with the quantized gradient? Do you perform the quantized gradient over the full-precision weight?

b. What kind of optimizer you used for the experiments?

c. How do you handle the precision of the error term (defined in WAGE paper [3])?

Overall, I think the paper is interesting and it would be worth being accepted after addressing the suggestions above.


[1] Alistarh, Dan, et al. "QSGD: Communication-efficient SGD via gradient quantization and encoding." Advances in Neural Information Processing Systems. 2017.

[2] Bernstein, Jeremy, et al. "signSGD with majority vote is communication efficient and fault tolerant." arXiv preprint arXiv:1810.05291 (2018).

[3] Wu, Shuang, et al. "Training and inference with integers in deep neural networks." arXiv preprint arXiv:1802.04680 (2018).

*************
I want to thank the authors for their detailed comments and explanations. I agree with (3) that a global floating representation is not required, but different settings of bitwidth may require additional data paths designed in the hardware (therefore a unified bitwidth for exponent and mantissa is recommended). Given revised related work and additional practical implications, I would like to champion this paper.

---

> ### Author Response · Authors · 2020-11-18
> **Authors response for Reviewer #3**
>
> 1) QSGD and signSGD compress the weight gradients to make distributed training in data parallelism more efficient. In our work, we compress the neural gradients  (gradients with respect to intermediate neural layer outputs) to reduce the computation resources and bandwidth (including in model parallelism). Following the reviewer's request we added QSGD and signSGD to the previous work and distinguish them from our work.
>
> 2) The compression of the neural gradients reduce the bandwidth, memory footprint and computational time. The bandwidth and memory footprint is reduced since in the backpropagation process, all the intermediate calculation are saved in the device memory before the calculation of next layers (usually cache memory can't contain all these intermediate calculation and it needs to be transfered to the SRAM).  Additionally, in distributed training of model parallelism, the neural gradients may form a bottleneck, as these gradients are required to transfer across devices (see recent paper by Facebook [1] explaining this problem in recommendation systems). Computational effort is also reduced, since the neural gradients are part of the calculation of next layer and weight gradients at backpropagation.
>
> As proposed by the reviewer we added a new paragraph to the introduction explaining this motivation.
>
> 3)  We would like to note that contrary to reviewer's comment, real hardware does not require "global floating representation across all datasets" and, in fact, may use different precision even within a single model ("mixed precision" - see https://developer.nvidia.com/automatic-mixed-precision for example). Nevertheless, our simulations show a considerable overlap between different data sets, which allows defining a single bit allocation representation that can target several data sets simultaneously (e.g., the ideal sign-exponent-mantissa bit allocations for the FP8 format is 1-5-2 for both CIFAR100 and ImageNet). Moreover, even without this overlap, since we still have to choose some format for the hardware device, we believe it is useful to know which formats would be most beneficial for which combinations of precision/dataset/model. Lastly, our analysis can also be useful the other way around. Given a pre-defined sign-exponent-mantissa bit allocation, we can find the best dynamic range (i.e., $\sigma$) for the gradients' operation. With that in mind, future work can study how gradients can be brought to this operating point (e.g., by manipulating training hyperparameters, such as the learning-rate), hence minimizing the quantization noise for an already specified floating-point format.
>
> 4) As we explained in (1) above, in this work we focus on quantization/pruning of neural gradients rather than weight gradient (as is done in QSGD and signSGD). In order to calculate the weight update, we simply multiply the quantized/pruned neural gradient with the input to the corresponding layer.
>
> 5) We used SGD with momentum, as mentioned in section A.12 (in the appendix), along with the other details of the implementation. Additionally, the code to reproduce all the experiments is available in the supplementary material.
>
> 6) The error term (as defined in WAGE paper) is exactly what we called neural gradient. In WAGE, authors quantize it to INT8 format and use a FP32 scaling factor (maximum of the tensor). We propose a more efficient scaling, combined with FP quantization.

---

### Official Review · AnonReviewer1 · 2020-10-29
**Interesting observation led to nice computation savings, but need more evaluation**

**Rating:** 6
**Confidence:** 4

**Review:**


This work proposed a very interesting idea that the back-propagated errors have log-normal distributions. The authors could extend this intriguing observation into computation efficient algorithms; reduced-precision floating-point quantization or the pruning of the back-prop error, which are very interesting. The authors also provide details of derivations as well as the data analysis along with several small ideas (in Appendix) to strongly support their ideas.

The biggest shortcoming would be the relatively shallow evaluation of the proposed method. In particular, since the main ideas of this work relies on a very strong assumption that the back-propagated errors are log-normal distributions, (and I guess it would not be possible to analytically prove this assumption) it would be necessary to provide thorough investigations on various deep learning applications to validate this assumption. Table 1 seems to be a good start, but it is a little disappointing to see that a very limited set of image classification models are tested with the proposed method in Table 3 and Fig 5.

In fact, one concern about the proposed method is that the back-prop error might evolve drastically throughout the training periods, and there is no guarantee for it to maintain its shape. Therefore, it would be critical for the authors to convince that the proposed method work in a wide range of applications (at least the kinds shown in Table 1, since this table claims that the evaluated models here show log-normal shapes).

In summary, this work seems to be interesting, but it should be supported by a much more thorough evaluation.

---

> ### Author Response · Authors · 2020-11-18
> **Authors response for Reviewer #1**
>
> 1) Following the reviewer's request, we now evaluate the pruning method on two additional models (MobileNetV2 and BERT). Fig A.24 shows that the lognormal distribution is key to accurately setting the required sparsity level.  Together with the three image classification tasks we already had, we now demonstrate the pruning method's accuracy and efficiency on 5 different models. For quantization, we have already evaluated our methods on 3 different models using 2 datasets and a diverse set of FP configurations, resulting in a mix of 28 full training runs (see Table 3). We will add more experiments until the camera ready, as these take more time.
>
> 2)  We added a few more experiments to evaluate the empirical distribution of neural gradients. We now demonstrate the validity of the log-normal distribution throughout training at different epochs and different models, as shown in Table A2 and Fig A.3 (see appendix). As the reviewer can see, the distribution shape is stable across the training.
>
> 3) We agree with the reviewer that it is probably hard to prove analytically the lognormal distribution of the neural gradients. However, for intuition of the lognormal distribution behaviour, please look at answer (1) for Reviewer #4.

---

### Official Review · AnonReviewer4 · 2020-10-30
**new insights on gradient quantization**

**Rating:** 8
**Confidence:** 4

**Review:**

The authors show that gradients from backpropagation are lognormally distributed empirically across several popular architectures like (Bert, ResNet18, MobileNetV2, VGG16, DenseNet121) over widely used datasets (CoLa, MRPC, ImageNet and CIFAR100). Using this result, they propose schemes for gradient quantization and gradient pruning that are theoretically principled and outperform existing methods in the literature.

I was very impressed by the principled nature of the authors' approach. Proving the lognormal distribution using a KS test and using it elegantly to find analytical formulations for optimal bit division in quantization and threshold parameter for sparsity in gradient pruning was masterful.

* In the appendix, the KS test shows that logLaplace distribution is also a good fit for the gradient distribution. Can the authors provide any intuition as to why a log normal might be better?

* Clearly for pruning, logNormal distribution seems to work better than logLaplace. It’d be interesting to see the results of using LogLaplace distribution for quantization and verify that it yields worse results there too.

* Can the authors elaborate on time/memory complexity, and how well their methods made improvements for training time?

---

> ### Author Response · Authors · 2020-11-18
> **Authors response for Reviewer #4**
>
>
>
> 1) This is a very interesting question, but we so far we did not find a simple or "intuitive" answer. From the central limit theorem, normal distributions universally arise from the sum of many random variables, while lognormal distributions universally arise as the product of many random variables. This might suggest that, as we backpropgate the neural gradients through the network, these gradients have a few dominant paths, so most significant operations are products, rather than summations (i.e., so along these paths: effective depth >> effective width). However, we suspect this might be only a part of a more nuanced picture.
>
> 2)  In order to show that the loglaplace distribution assumption yields worse results in quantization, we added  Fig A.1.b to the appendix showing the FP format obtained by solving Eq.3 with loglaplace distribution (Similar to Fig3b and Fig3c). Notice this leads to sub-optimal format - for example in Cifar100 FP6 format, loglaplace assumption leads to 1-5-0 as the optimal format, which is sub-optimal as shown in Table 3.
>
> 3) The overhead of the proposed method in pruning is very small, as we elaborated in section A.7 --- we calculate the threshold only once per epoch, and even then, this adds less than 5{\%} to that single iteration. In all other iterations, the overhead is negligible. We noticed a similar overhead to find the per-layer scaling for quantization. In total, both methods add minimal overhead to the training process.
> The compression of the neural gradients reduce the bandwidth, memory footprint and computational time. The bandwidth and memory footprint is reduced since in the backpropagation process, all the intermediate calculation are saved in the device memory before the calculation of next layers (usually cache memory can’t contain all these intermediate calculation and it needs to be transfered to the SRAM). Additionally, in distributed training of model parallelism, the neural gradients may form a bottleneck, as these gradients are required to1 transfer across devices. Computational effort is also reduced, since the neural gradients are part of the calculation of next layer and weight gradients at backpropagation

---

### Author Response · Authors · 2020-11-18
**New revision uploaded**

We thank all the reviewers for their helpful feedback and remarks.  We uploaded a new revision of the paper to address these remarks. For additional details, see the answer for each of your concerns. Please let us know if there is any additional comment.

---

### Decision · Program_Chairs · 2021-01-07
**Final Decision**

**Decision:**

Accept (Poster)

**Comment:**

This work makes the observation that gradients in neural network training are approximately distributed according to a log-normal distribution. This observation is then used to compress and sparsify the gradients, which can be useful in distributed optimization of neural nets. The reviewers indicate that this contribution is novel and useful and they do not find any major issues with the presented work. I recommend accepting the paper for a poster presentation.